# Investigating the Association Between Citrus Huanglongbing and Chlorophyll Content Using Hyperspectral Detection

**DOI:** 10.3390/s25237292

**Published:** 2025-11-30

**Authors:** Shiqing Dou, Minglan Li, Xiangqian Qi, Yichang Hou, Shixin Yuan, Yaqin Song, Zhengmin Mei, Genhong Qi

**Affiliations:** 1College of Geomatics and Geoinformation, Guilin University of Technology, Guilin 541006, China; 2Guangxi Ecological Spatiotemporal Big Data Perception Service Laboratory, Guilin 541004, China; 3School of Resource Engineering, Longyan University, Fujian 364012, China; 4Guangxi Key Laboratory of Germplasm Innovation and Utilization of Specialty Commercial Crops in North Guangxi, Guangxi Citrus Breeding and Cultivation Technology Innovation Center, Guangxi Academy of Specialty Crops, Guilin 541004, China

**Keywords:** leaf chlorophyll content, Huanglongbing, feature selection, machine learning, spectral reflectance

## Abstract

Huanglongbing (HLB) poses a severe threat to the sustainable citrus industry, causing significant alterations in the spectral reflectance and leaf chlorophyll content (LCC) of citrus leaves. This study investigates the quantitative relationship between spectral characteristics and LCC for the early detection of HLB in Mianju mandarin cultivars. We analyzed hyperspectral data from healthy and HLB-infected leaves, employing the least absolute shrinkage and selection operator (LASSO) method and spectral indices to select chlorophyll characteristic bands, and several machine learning models were used to estimate the LCC. The results indicate that: (1) HLB-infected leaves exhibit significantly different spectral reflectance, characterized by a distinct “blueshift of the red edge”; (2) a greater proportion of characteristic bands for HLB-infected leaves were located in the near-infrared region compared to healthy leaves; and (3) the LASSO-PLSR model demonstrated high predictive accuracy for LCC estimation—for healthy leaves (Rv^2^ = 0.956, RMSEv = 0.675) and for HLB-infected leaves (Rv^2^ = 0.816, RMSEv = 4.614)—with performance being notably superior for healthy leaves (Rv^2^ difference of +0.146). This research establishes a systematic quantification between hyperspectral and chlorophyll content, suggesting that hyperspectral-based LCC estimation can serve as a reliable indirect indicator for the early detection of HLB, with substantial practical application potential.

## 1. Introduction

Citrus is among the most important global cash crops. As a major citrus-growing nation, China ranks first in terms of global production and exports [1,2]. Huanglongbing (HLB), also known as citrus greening disease or citrus yellowing disease, has been termed the “cancer” of citrus because of its rapid spread, severe damage, high mortality rate, and lack of viable therapies to date. HLB threatens the long-term growth of the citrus industry [3,4].

When citrus trees become infected with HLB, the pathogen colonizes the phloem tissue within the leaf veins, disrupting the long-distance transport of photosynthetic products. This disruption results in the disintegration of chloroplast structures and a rapid decrease in chlorophyll content, causing mottled yellowing symptoms on leaves and damaging citrus growth [5,6]. Early research revealed that when healthy citrus trees were infected with HLB, their photosynthetic processes began to decrease. Significant decreases in chlorophyll a, chlorophyll b, and carotenoid content were already visible during the latent period when citrus trees infected with HLB did not show noticeable symptoms [7]. As Shi et al. [8] reported, the chlorophyll a and chlorophyll b contents were significantly lower in symptomatic HLB-infected leaves from field-grown trees compared to healthy leaves. This finding confirms the strong correlation between citrus HLB and chlorophyll indicators. Therefore, as a direct indicator of photosynthetic capacity, the precise estimation of chlorophyll content is essential for the early detection of HLB infection. These physiological changes in citrus trees further alter leaf optical properties, particularly in the visible spectrum region (400–700 nm). These include changes in absorption characteristics in the blue spectrum region (approximately 450 nm) and red spectrum region (approximately 680 nm), as well as reflectance anomalies in the near-infrared spectrum [9]. Before HLB symptoms develop, changes in chlorophyll content and impaired photosynthetic mechanisms can be detected through spectral alterations [10]. Therefore, the use of hyperspectral technology to thoroughly investigate HLB-induced leaf spectral changes and their coupling mechanisms with chlorophyll content is crucial for advancing hyperspectral-based early nondestructive detection of HLB, as well as for precision management and disease detection in citrus orchards.

Recently, the advantages of hyperspectral technology for estimating citrus chlorophyll content have been well reported in numerous studies [11,12,13,14]. Li et al. [13], Ansar Ali et al. [15,16], Xiao et al. [17] and Peng et al. [18] have successfully estimated the chlorophyll content in citrus plants at the landscape, canopy and leaf scales, respectively. Yue et al. [19] and Li et al. [11] investigated the effects of distinct growth stages on the estimation of chlorophyll content by obtaining hyperspectral data from citrus leaves at various growth stages to predict the chlorophyll content over the course of the growth period and at a single time point, respectively. Few studies have created leaf chlorophyll content estimation models on the basis of the spectra of citrus leaves infected with HLB, although hyperspectral technology is becoming increasingly ready for use in estimating the chlorophyll content of citrus leaves [20,21]. The association between chlorophyll content and spectral response in HLB-infected citrus leaves remains poorly systematically quantified [22,23]. Therefore, investigating the differences in chlorophyll-sensitive bands between HLB-infected leaves and healthy leaves and thus exploring the relationship between HLB infection and chlorophyll content is important for the development of early diagnostic indicators and the optimization of chlorophyll estimation models.

Hyperspectral reflectance is currently being used by both domestic and foreign researchers to conduct in-depth analyses of the physiological and biochemical data of various crops [24,25,26]. Creating estimation models based on feature variables to calculate the amount of chlorophyll in crop leaves has been the main focus of many studies [27,28]. However, the performance of feature selection methods can vary significantly across different datasets. Therefore, to construct effective chlorophyll estimation models for HLB-infected and healthy leaves, this study evaluates and compares several widely applied algorithms, including the least absolute shrinkage and selection operator (LASSO) [29,30,31,32], competitive adaptive reweighting selection (CARS) [33], recursive feature elimination (RFE) [34,35,36], and spectral index construction [37,38,39]. These methods effectively eliminate redundant information and improve model interpretability and generalizability.

Given the spectral differences between healthy and HLB-infected leaves, combining hyperspectral data with machine learning to estimate chlorophyll content is a feasible approach for early HLB identification. To systematically validate this approach and explore the essential mechanisms linking HLB infection to chlorophyll content, the objectives of this study are as follows:(1)The relationships between chlorophyll content and hyperspectral responses in healthy and HLB-infected citrus leaves will be compared and analyzed;(2)Optimal feature selection methods will be explored, spectral band differences between healthy and HLB-infected leaves will be compared, the predictive performance of feature bands will be evaluated, and optimal chlorophyll content estimation models for healthy and HLB-infected leaves will be established;(3)The coupling mechanism between HLB infection and changes in chlorophyll content in leaves will be investigated via a comparison of LCC estimation models for healthy and HLB-infected leaves.

## 2. Materials and Methods

### 2.1. Study Area

The primary data used in this study were collected from the citrus experimental base in Xinyu Shang, Sanjie town, Lingchuan County, Guilin city, Guangxi (Figure 1a). This area has a subtropical monsoon climate characterized by abundant sunlight, plentiful rainfall, and rainfall and heat occurring during the same period. Mianju citrus trees were cultivated primarily in the study area, some of which were infected with HLB. Considering factors such as tree species and age, field sampling was conducted on 19 November 2024. The health status of the trees was visually identified by field experts. A total of 320 Mianju leaf samples were randomly collected: 120 samples from trees suspected of HLB infection and 200 samples from apparently healthy trees. After the hyperspectral and LCC data were obtained, all the samples were sent to the Guangxi Academy of Specialty Crops for definitive HLB detection.

Murcott citrus data was introduced to evaluate the predictive capability and generalization ability of the machine learning models selected by the research institute across different varieties. The data were collected from the citrus practice base in Tangyuan Village, Tanxia Town, Lingchuan County, Guilin City, Guangxi (Figure 1b), with sampling performed on 9 November 2025. The Murcott variety was cultivated in this area. A total of 270 leaf samples were collected, including 120 healthy samples and 150 HLB-infected samples (with obvious symptoms). The health status of samples in this dataset was determined by domain experts based on visual field observations, without subsequent laboratory testing for HLB.

### 2.2. Hyperspectral Data Acquisition and Spectral Preprocessing

A portable analytical spectral device (ASD FieldSpec 4 Stand Res, Boulder, CO, USA) was used in this study to obtain the raw hyperspectral data of the leaf samples. The spectra were produced at a resampled interval of 1 nm, with the spectral range between 350 and 2500 nm. The ASD equipment was calibrated using the calibration white panel in the portable leaf clip after being powered on for more than fifteen minutes to warm up before measurement. The instrument was recalibrated every fifteen minutes while the data was being collected. The adaxial (upper) surface of the leaf was oriented towards the light source when it was inserted into the handheld leaf clip for each leaf spectrum measurement. Citrus leaves are typical heterophyllous leaves, with the mid-leaf region serving as the mature functional zone exhibiting relatively uniform pigment distribution. To ensure data representativeness and consistency, the study selected the central region of citrus leaves (avoiding the main vein) as the spectral measurement area. The measurement area had a diameter of approximately 1.5 cm (consistent with the effective measurement window size of the handheld leaf holder) (Figure 1d). Ten spectral curves were preserved from the measurement area, and the average of these ten spectral curves was taken as the hyperspectral data for this leaf sample.

Appropriate spectral preprocessing mitigates instrumental noise, reshapes absorption features and stabilizes baseline drift, increasing the predictive capacity of subsequent modeling. Accordingly, to improve data quality, the raw spectral data in this study were preprocessed by integrating the fractional order derivative (FOD) with each of the following four methods: raw spectra (RAW), Savitzky–Golay smoothing (SG, with a window size of 21 points and a 3rd-order polynomial), standard normal variate (SNV), and multiplicative scatter correction (MSC).

### 2.3. Chlorophyll Acquisition and TR-PCDR Detection

In November 2024, apparently healthy and suspected HLB-infected leaves were randomly collected from ‘Mianju’ citrus trees at the Citrus Experimental Station. The leaves were placed in sealed plastic bags and brought back to the laboratory. The chlorophyll content in the citrus leaves was measured by using a SPAD-502 chlorophyll meter (Konica Minolta, Tokyo, Japan). The soil and plant analysis development (SPAD) value can be used to determine the relative chlorophyll content, with higher values indicating greater chlorophyll concentrations. Numerous studies have reported a high correlation between SPAD values and chlorophyll content in fruit tree leaves [40]. Among them, Ye et al. [41] found that there was a significant correlation between the SPAD value and chlorophyll content in apple leaves (r = 0.9318 (*p* < 0.001)). Therefore, in this study, SPAD values were used as indicators of leaf chlorophyll content (LCC). For each leaf, three measurement points were selected away from the midrib using a chlorophyll meter. The average of these three readings was used as the LCC (Figure 1f).

To increase the reliability of the experimental data, after the hyperspectral data and SPAD values were obtained, the leaf samples were sequentially stored in resealed bags for preservation and labeling. These 320 leaf samples were subsequently sent to the Guangxi Academy of Specialty Crops for HLB diagnosis using tandem repeat-based polymerase chain displacement reaction (TR-PCDR) technology. Midribs were excised from each leaf sample and ground in liquid nitrogen. Total genomic DNA was extracted using a modified cetyltrimethylammonium bromide method from 100 mg of the ground midrib tissue, as described previously [42]. The DNA pellet was dissolved in 500 μL of Tris-EDTA buffer (10 mM Tris-HCl and 1 mM EDTA, pH 8.0) and stored at −20 °C for later use.TR-PCDR was carried out on an S1000 thermal cycler (Bio-Rad, Hercules, CA, USA) using 20 μL of reaction mixture containing 1× SD polymerase reaction buffer, 3 mM MgCl2, 375 μM each dNTP, 0.5 μM TR2-PCDR-F, 0.5 μM TR2- PCDR-1R, 4 U of SD Polymerase (BIORON, Ludwigshafen, Germany) [43], and 1 μL of diluted DNA. The parameters for amplification were as follows: 92 °C for 2 min, followed by 35 cycles of 92 °C for 30 s and 62 °C for 2 min, and a final incubation of 10 min at 62 °C. Each TR-PCDR run contained one negative control and one positive control. Amplification products were separated by electrophoresis in 1.5% agarose gels and visualized by staining with 4S Red Plus Nucleic Acid Stain (Sangon Biotech, Shanghai, China) (Figure 1e).The sensitive level of TR-PCDR was proven to be 100× higher than conventional PCR and similar to real-time PCR [44]. After TR-PCDR detection, samples inconsistent with the results of the preliminary field assessments were discarded, resulting in a final dataset comprising 100 healthy leaves and 105 HLB-infected leaves (Table 1).

### 2.4. Exploring the Correlations Between HLB Citrus Leaves and Chlorophyll Contents Using Hyperspectral Data

#### 2.4.1. Comparative Analysis of the Chlorophyll and Hyperspectral Responses for Healthy and HLB-Infected Citrus Leaves

In this study, color mapping and Pearson correlation analysis were employed to explore the relationship between the chlorophyll content and hyperspectral reflectance in healthy and diseased leaves. Initially, color mapping was used to visually represent the hyperspectral data from both healthy and HLB-infected leaves, allowing for the intuitive identification of differences in spectral characteristics and trends between the two leaf types. Pearson correlation analysis was further employed to calculate the correlation coefficients between the hyperspectral reflectance and LCC for healthy and HLB-infected leaves. The association between spectral alterations caused by HLB disease and chlorophyll content was examined by comparing the variations in correlation coefficients between the spectral properties of the two leaf types and LCC.

#### 2.4.2. Comparative Analysis of the Feature Bands Between Healthy and HLB-Infected Leaves

Hyperspectral data have high resolution and rich information content. Directly using all spectral bands as variables in estimation models inevitably results in the incorporation of substantial amounts of redundant information, which may decrease the predictive ability of the developed models. Thus, LASSO and spectral indices were employed in this study to select feature bands from full-spectrum data. The distribution of specific feature bands across spectral regions (such as the visible spectrum, red edge, and near-infrared) in both healthy and HLB-infected leaves was statistically examined in this work. The effect of HLB disease on leaf spectral features and chlorophyll response patterns was further revealed by a comprehensive comparison of the spatial distribution differences in chlorophyll-sensitive bands between the two groups.

The LASSO algorithm performs well with high-dimensional datasets and nonstationary distributed data. The algorithm integrates an L1 regularization term into the loss function, shrinking several feature coefficients to zero to achieve optimal feature selection [45]. Additionally, CARS and RFE will be employed for feature selection and compared with LASSO. Ten spectral indices (RI, DI, NDVI, RDVI, ARI, NLI, mSR, mNDI, TVI, SAVI) listed in Table 2 were selected for feature selection of the Mianju leaf spectral data. For each of the 10 spectral indices, the specific wavelength combination (R_i_, R_j_) that demonstrated the highest absolute Pearson correlation with LCC was identified through a band-by-band calculation process. These optimized versions of the indices were then used for subsequent model development. For the indices mSR, mNDI, and TVI, we adhered to their standard definitions, which prescribe fixed wavelengths at 445 nm and 500 nm (R_445_ and R_500_)—a choice validated in prior studies [46]. Consequently, in our optimization, these bands remained fixed, and only the complementary bands (R_i_ and R_j_) were variable. In contrast, both wavelengths for the remaining indices were fully optimized.

#### 2.4.3. Comparison of Models for Estimating Chlorophyll Contents in Healthy and HLB-Infected Citrus Leaves

In this study, six theoretically representative machine learning algorithms were selected to construct estimation models for citrus LCC: partial least squares regression (PLSR), support vector regression (SVR), random forest (RF), adaptive boosting (AdaBoost), extreme gradient boosting (XGBoost), and categorical boosting (CatBoost). All the models were implemented using Scikit-learn and the relevant optimization libraries. To ensure robust model development and prevent overfitting, hyperparameter optimization was conducted via grid search coupled with 5-fold cross-validation on the training set. Specifically, for each grid search, the training set was partitioned into 5 folds; the model was iteratively trained on 4 folds and validated on the remaining 1 fold to identify the optimal hyperparameter set. To further elucidate the variation in LCC between healthy and HLB-infected leaves, LCC estimation models for healthy and HLB-infected leaves were constructed in this study using the optimized algorithm described above. By comparing the optimal performance and characteristic bands of the two-leaf models, this study systematically quantified and demonstrated LCC changes between healthy and HLB-infected leaves across three dimensions: prediction accuracy, spectral response mechanisms, and quantitative estimation results.

To ensure a rigorous and unbiased evaluation of the model’s generalization ability, the entire dataset was randomly split into a training set and an independent test set at a ratio of 8:2. This test set was strictly held out from the entire model training and hyperparameter optimization process and was not involved in any way. Model performance was evaluated with the root-mean-square error (RMSE), coefficient of determination (R^2^) and residual prediction deviation (RPD). R^2^ represents the ability of a model to explain variance in the dependent variable, with values of [0, 1]. Values closer to 1 indicate excellent explanatory power. RMSE serves as an indicator of model accuracy and estimation capability; lower values denote stronger precision and estimation ability. RPD is a critical metric for assessing the predictive capacity of a model. The predictive capability is interpreted as follows: an RPD less than 1.4 indicates poor performance; an RPD between 1.4 and 2.0 suggests acceptable predictive ability; an RPD ranging from 2.0 to 2.5 denotes reliable estimations; and an RPD exceeding 2.5 signifies strong model stability and accuracy and is suitable for practical application [49]. Parameter values are denoted by the subscript letters c and v for the training set and test set, respectively.(1)$$\mathrm{RMSE}=\sqrt{\frac{1}{n}\sum_{i=1}^{n}{\left(y_{i}-{\hat{y}}_{i}\right)}^{2}}\;$$(2)$$R^{2}=1-\frac{\sum_{i=1}^{n}{\left(y_{i}-{\hat{y}}_{i}\right)}^{2}}{\sum_{i=1}^{n}{\left(y_{i}-{\overset{-}{y}}_{i}\right)}^{2}}$$(3)$$\mathrm{RPD}=\frac{1}{\sqrt{1-R^{2}}}$$
where n represents the sample size, $y_{i}$ denotes the actual chlorophyll content, ${\hat{y}}_{i}$ indicates the predicted chlorophyll content, and ${\overset{-}{y}}_{i}$ represents the average of the actual chlorophyll content across all the samples.

## 3. Results and Analysis

### 3.1. Comparative Analysis of the Chlorophyll and Hyperspectral Responses for Healthy and HLB-Infected Mianju Citrus Leaves

Hyperspectral data of Mianju leaves within the 350–2500 nm range were obtained. To explore the relationship between the leaf spectra and LCC in both HLB-infected and healthy leaves, the obtained hyperspectral reflectance was presented using color mapping. The SPAD values were normalized and mapped to a color gradient across the spectral curves, and distinct colors were assigned to spectra of different chlorophyll content levels. Figure 2 presents the color-mapped spectra of HLB-infected and healthy Mianju leaves in relation to their LCC. A distinct trend can be observed in the 500–750 nm range, particularly near the minor reflectance peaks: lower chlorophyll content corresponds to higher reflectance at the green peak (~550 nm) and the red valley (~680 nm). This result is likely because HLB-induced chlorophyll degradation reduces the absorption of red light and increases the reflection of green light. A steep slope, known as the red edge, is observed in the 680–750 nm band. As shown in Figure 2a, the red edge position in HLB-infected leaves exhibited a distinct shift towards shorter wavelengths, which is referred to as the “blueshift of the red edge”.

The analysis of mean spectral differences between healthy and HLB-infected leaves revealed the most pronounced variations in the visible and red-edge regions, with reflectance differences reaching 0.144 at 544 nm and 0.209 at 710 nm, respectively (Figure 3a). These spectral disparities were further elucidated by the correlation analysis between spectral reflectance and LCC (Figure 3b). Although both healthy and infected leaves exhibited generally consistent correlation patterns with LCC, HLB-infected leaves demonstrated stronger negative correlations in the visible region, potentially attributable to chlorophyll degradation enhancing the reflectance-SPAD relationship. Conversely, the red-edge region, which showed the greatest spectral differences, displayed weakened correlations approaching zero, indicating the combined influences of chlorophyll content and leaf structural factors in this spectral domain.

### 3.2. Comparative Analysis of Combinations of Spectral Preprocessing for Healthy and HLB-Infected Mianju Citrus Leaves

Spectral preprocessing was performed by combining RAW, SG, SNV, and MSC with FOD, and the performance of different preprocessing combinations in predicting LCC across the full wavelength range was evaluated using PLSR models. Figure 4 shows the R^2^ values for LCC estimation based on the PLSR model for healthy and HLB-infected Mianju citrus leaves with different preprocessing combinations.

Figure 4 shows the different effects of different spectral preprocessing combinations on the performance of full-spectrum PLSR models. With respect to healthy leaves (Figure 4a), all the preprocessing methods produced models with better predictive performance across FOD orders of 0.2–0.5 than those using raw spectra did. The best-performing combination was SNV-FOD0.4, whose Rv^2^ was 0.803. With respect to HLB-infected leaves (Figure 4b), the optimal test set performance (Rv^2^ = 0.725) was obtained with MSC or SNV preprocessing only, a significant improvement over the model based on raw spectra (Rv^2^ = 0.667). Overall, the predictive accuracy of all the preprocessing combinations on HLB-infected Mianju citrus leaves was markedly lower than for healthy leaves, indicating that HLB infection complicates the response mechanism between the leaf spectra and LCC.

### 3.3. Comparative Analysis of the LASSO Feature Bands for Healthy and HLB-Infected Mianju Citrus Leaves

In this study, LASSO was employed to select feature bands from spectra preprocessed using the RAW-FOD, SNV-FOD, SG-FOD, and MSC-FOD methods. The performance of different feature bands was evaluated on the basis of PLSR models. Two classical feature selection algorithms—RFE and CARS—were employed for comparison with LASSO. Table 3 lists only the optimal FOD order for the best predictive performance of the three feature selection algorithms with different preprocessing combinations.

As show in Table 3, the LASSO algorithm selected fewer feature bands than the CARS and RFE approaches did while achieving better prediction performance in both the healthy and HLB-infected Mianju leaf datasets. The feature combination obtained from the raw spectra with 1.0-order FOD preprocessing using LASSO (RAW-FOD1.0-LASSO) yielded the best prediction results in the PLSR model for healthy leaves, with an RMSEv of 0.81 and an Rv^2^ of 0.94. The RAW-FOD0.9-LASSO feature combination had the best predictive performance for HLB-infected leaves, with an RMSEv of 4.70 and an Rv^2^ of 0.82. LASSO had the most stable and optimal performance for the LCC prediction task after a thorough assessment of the three feature selection methods across various preprocessing methods.

To further investigate the differences in distribution for the optimal feature bands between healthy and HLB-infected Mianju citrus leaves, the proportion of feature bands selected by LASSO across different spectral regions with various preprocessing combinations was analyzed (Figure 5). Spectral regions were defined as the visible region (VIS, 400–680 nm), red edge region (RED, 680–750 nm), near-infrared region (NIR, 750–1300 nm), and shortwave infrared region (SWIR, 1300–2500 nm).

As observed in Figure 5, the chlorophyll-sensitive feature bands selected by LASSO for healthy leaves with different pretreatment combinations were predominantly located in the visible (VIS), near-infrared (NIR), and shortwave infrared (SWIR) regions. Among these regions, the proportion of selected bands in the SWIR region was generally greater than that in the other three regions. The feature bands selected for HLB-infected leaves with various pretreatment combinations were slightly more concentrated in the VIS and SWIR regions than in the red-edge and NIR regions. A comparison of the feature bands selected by LASSO from healthy and HLB-infected leaves revealed that the proportion of feature bands in the red edge region (680–750 nm) was significantly greater in HLB-infected leaves than in healthy leaves. This difference was observed across multiple preprocessing methods (particularly RAW and SG) at various FOD orders.

### 3.4. Comparative Analysis of the Optimal Spectral Indices for Healthy and HLB-Infected Mianju Citrus Leaves

In this study, 10 spectral indices were calculated using spectral index formulas. The correlation coefficient matrix diagram was computed and plotted for each spectral index in relation to LCC using Pearson’s correlation coefficient approach. From each correlation coefficient matrix diagram, the optimal band combination with the highest correlation coefficient was subsequently chosen. As shown in Figure 6, the transition from blue to red indicates that the correlation between spectral indices and LCC for HLB-infected and healthy leaves shifts from a high negative correlation to a high positive correlation.

As shown in Figure 6, the spectral indices constructed within the 350–850 nm range for both HLB-infected and healthy leaves generally exhibited higher correlation coefficients with LCC than those derived from other spectral regions did. For the same spectral index, the correlation with LCC was consistently greater for HLB-infected leaves than for healthy leaves. As indicated in Table 4, all the optimal spectral indices were significantly correlated with LCC. The optimal index for HLB-infected leaves (difference index, DI) yielded a higher correlation coefficient than the optimal index for healthy leaves (triangular vegetation index, TVI).

Observation of the optimal spectral index band combinations for HLB-infected and healthy leaves in Table 4 revealed significant differences in the positions of their constituent bands. The band combinations for the HLB-infected leaf spectral index were predominantly distributed in the NIR and red edge regions. In contrast, the band combinations for the healthy leaf spectral index were dispersed across multiple regions, including the blue–green, red-edge to NIR, and NIR plateau regions. This difference may be attributed to the degradation of chlorophyll and the disruption of leaf tissue structure caused by HLB infection. These pathological changes shift the sensitive spectral regions of LCC from being broadly distributed in healthy leaves to becoming concentrated in the red-edge and NIR regions (particularly 690–830 nm).

In this study, PLSR models were employed to estimate LCC using both the spectral index (SI) and the constituent band of the spectral index (SI_CB). To further avoid informational redundancy, PLS was employed to iteratively eliminate variables from the SI and SI_CB feature variables, retaining the optimal feature combinations (denoted as SI_PLS and SI_CB_PLS, respectively). Table 5 presents the predictive performance of the feature combinations for both healthy and HLB-infected leaves. Overall, compared with using the full-range raw spectra directly, using spectral indices for LCC prediction resulted in a comparable improvement in model performance and a reduction in the number of feature variables.

Table 5 shows that the percentage of SI_CB_PLS feature bands located in the near-infrared (NIR) region was 80% for HLB-infected leaves, which was 46.6% greater than that observed for healthy leaves. As shown in Figure 7 and Figure 8, the spectral index-selected feature bands (SI_BC) employed PLSR for LCC estimation, yielding model performance superior to that of directly use of spectral indices (SI). Furthermore, after feature selection was performed again using PLS, the model performance improved, outperforming the LCC estimation achieved by PLSR using the full-band raw spectra. Moreover, the performance of the model considerably improved when feature selection was performed again.

### 3.5. Comparison of the Chlorophyll Estimation Models for Healthy and HLB-Infected Mianju Citrus Leaves

To systematically evaluate the robustness and representativeness of the selected feature variables, six mainstream machine learning regression models (PLSR, SVR, RF, AdaBoost, CatBoost, and XGBoost) were used to predict LCC. Using the corresponding LASSO-selected feature bands and the SI_CB_PLS characteristics shown in Table 6, this assessment was conducted independently for both healthy and HLB-infected Mianju leaves.

The performance of the LCC prediction models developed from the optimal feature bands for both healthy and HLB-infected Mianju citrus leaves is shown in Figure 9, Figure 10, Figure 11 and Figure 12.With respect to healthy and HLB-infected Mianju leaves, the feature bands selected by LASSO yielded excellent predictive performance across all six machine learning models compared with those selected by spectral indices (Table 7). With respect to the LCC prediction for healthy leaves, compared with the RF, AdaBoost, and CatBoost models, both feature bands (RAW-FOD1.0-LASSO and SI_CB_PLS) performed markedly better with the PLSR, XGBoost, and SVR models. Specifically, the best prediction performance was achieved for the RAW-FOD1.0-LASSO feature bands when the PLSR model was used (Rv^2^ = 0.956, RMSEv = 0.675, RPD = 4.767). With respect to HLB-infected leaves, the predictive performance of both characteristic bands was relatively stable across all six models. The RAW-FOD0.9-LASSO feature bands performed best with the PLSR model (Rv^2^ = 0.816, RMSEv = 4.614, RPD = 2.331). Notably, the LCC estimation models for HLB-infected leaves exhibited greater robustness, as observed in the narrower range of Rv^2^ values (0.737–0.816) for RAW-FOD0.9-LASSO across the models. Conversely, the LCC estimation model performance for healthy leaves was highly variable, as exhibited by the same feature set (RAW-FOD1.0-LASSO), yielding an Rv^2^ of 0.595 with CatBoost but an Rv^2^ of 0.956 with PLSR.

### 3.6. Estimation of LCC Using LASSO-Coupled Machine Learning Models on an External Dataset

To evaluate the LCC estimation capability of the methodology integrating LASSO with six machine learning models (PLSR, SVR, RF, AdaBoost, CatBoost, XGBoost) on a distinct dataset, an external dataset from a different citrus variety (Murcott) was utilized for model training and validation. The healthy and HLB-infected leaf samples of the Murcott variety were first randomly split into training and testing sets at an 8:2 ratio. Subsequently, the LASSO algorithm was employed to select feature bands from the spectral data of healthy Murcott leaves preprocessed with 0.9-order FOD and HLB-infected Murcott leaves preprocessed with 1.0-order FOD. Finally, the six machine learning models were retrained using the health and HLB-affected feature bands from Mercot, respectively, to construct new LCC estimation models and evaluate their performance (Figure 13 and Figure 14).

As shown in Table 8, for healthy Murcott leaves, the PLSR, SVR, and XGBoost models achieved highly consistent and excellent prediction performance, with Rv^2^ values around 0.88 and RPD values all around or exceeding 2.9. This indicates that the combination of LASSO feature selection with these models can reliably capture the stable association between chlorophyll content and spectral response in healthy leaves across different varieties. For HLB-infected Murcott leaves, the PLSR model achieved an Rv^2^ of 0.730, outperforming the other five models. However, the Rv^2^ values of the estimation models for infected leaves were consistently lower than those for healthy leaves, which is likely attributable to the increased complexity in spectral response induced by the disease. Table 7 and Table 8 demonstrate that when the LASSO-coupled machine learning approach was applied to the Murcott dataset, PLSR, SVR, and XGBoost successfully yielded estimation models with stable and significant predictive capability for both healthy and HLB-infected leaves. These models maintained stable and reliable performance across both the Mianju and Murcott datasets, confirming their advantage, particularly for models with a generalized linear basis, in learning universal physical patterns from spectral data. It is noteworthy that the RF and AdaBoost models exhibited severe performance degradation on the new dataset. For HLB-infected leaves, the Rv^2^ of the LASSO-AdaBoost model dropped from 0.807 (Mianju dataset) to 0.473 (Murcott dataset). Furthermore, in estimation studies on both datasets, the LASSO-CatBoost model showed signs of overfitting on the training set (Rc^2^ > 0.99), while its validation performance (Rv^2^ in the range of [0.575–0.737]) was substantially lower than that of the robust models. This indicates that the CatBoost model is highly prone to overfitting and demonstrates poor generalization capability across both datasets.

## 4. Discussion

### 4.1. Effects of HLB Infection on the Spectral Characteristics of Citrus Leaves

Citrus leaves infected with HLB exhibit visual symptoms such as irregular mottled yellowing and varying degrees of leaf curling [50,51]. These leaves also consistently exhibit lower leaf chlorophyll content (LCC) than their healthy counterparts do [52]. Therefore, the rapid and accurate estimation of LCC is highly important for citrus HLB detection. Compared with those in healthy leaves, alterations in spectral reflectance in the near-infrared (NIR) region in HLB-infected leaves are primarily associated with changes in internal chemical composition and tissue structure, such as abnormal carbohydrate accumulation and phloem blockage [53]. HLB-infected leaves exhibit changes in spectral reflectance across both the visible and NIR regions because of chlorophyll loss and cellular structural damage. As observed in Figure 2, compared with those of healthy leaves, the spectra of HLB-infected citrus leaves exhibit a “blueshift of the red edge” and significantly higher reflectance in the green–blue region. These findings are consistent with those reported in previous studies [54,55]. The band closely associated with chlorophyll was present in greater proportions in the visible and NIR regions in HLB-infected leaves than in healthy leaves (Figure 3). These phenomena could be explained by the decreased LCC in HLB-infected leaves, which affects LCC estimates by increasing reflectance in the visible region and increasing spectral sensitivity to lower chlorophyll levels. Therefore, it is necessary to investigate the differences between the characteristic spectral bands of chlorophyll in HLB-infected leaves and those in healthy leaves to enable more accurate estimation of the chlorophyll content in HLB-infected leaves.

### 4.2. Linear Regression Analysis for Chlorophyll Estimation Using Feature Bands of Healthy and HLB-Infected Leaves

The selection of feature bands via LASSO and spectral indices for both healthy and HLB-infected leaves revealed that bands in the blue, green, and red-edge regions were most frequently selected, highlighting their critical role in LCC estimation [17,49]. Figure 5 shows the distribution of the feature bands selected by LASSO across different spectral regions for both healthy and diseased leaves and that HLB-infected leaves had a significantly greater proportion of feature bands in the red-edge region (680–750 nm) than healthy leaves did. Analysis of the optimal spectral indices in Table 6 also revealed a distinct pattern: for HLB-infected leaves, the optimal spectral indices were primarily composed of bands within the near-infrared region, whereas for healthy leaves, the distribution of bands constituting the optimal spectral indices exhibited multiregional dispersion characteristics, predominantly spanning the blue–green visible light region and the transition zone from the red edge to the near-infrared region (631–750 nm). These findings are consistent with those reported by Tang et al. [56] and Li et al. [12]. This could be because biochemical and biophysical variables affected by HLB infection alter the spectral signature of the leaf, enabling the efficient use of the spectral information in this area to forecast LCC in HLB-infected leaves. Thus, the spectral in the red edge region serves not only as a sensitive indicator of chlorophyll content in HLB-infected leaves but also as a potential biomarker for early disease identification, providing a theoretical basis for the field diagnosis of citrus HLB.

The predictive performance of the feature bands selected by LASSO and spectral indices, which are based on six machine learning models, is summarized in Table 7. Although spectral indices are commonly employed as model variables in chlorophyll estimation studies and have yielded satisfactory results [37,46,47,48], the feature bands selected by LASSO in this study were highly accurate across all six machine learning models in estimating LCC. LASSO not only facilitates regression analysis but also automatically identifies the most influential features [45,57]. As shown in Figure 5, the feature bands selected by LASSO were distributed across a broader spectral range, enabling a more comprehensive set of chlorophyll-sensitive information from the hyperspectral data. This wider distribution likely enabled a more comprehensive set of chlorophyll-sensitive information from the hyperspectral data. In contrast, the bands constituting the spectral indices were often closely located and may have omitted some representative wavelengths, resulting in their slightly inferior performance for citrus LCC estimation compared with the LASSO-selected features.

The LCC prediction performance of six machine learning regression models (PLSR, SVR, RF, AdaBoost, CatBoost, XGBoost) was compared using the selected feature bands from both Mianju and Murcott for healthy and HLB-infected leaves (Table 7 and Table 8). The PLSR model demonstrated excellent performance for both healthy and HLB-infected leaves. This superiority could be attributed to the relatively small dataset size and the approximately linear relationship between the feature bands and LCC, as PLSR is well-suited for small-scale datasets and exhibits strong stability and robustness in handling linear regression problems [58,59]. Notably, the modeling approach incorporating LASSO feature selection revealed varying generalization capabilities across different models. PLSR, SVR, and XGBoost successfully established stable and reliable LCC estimation models on both variety-specific datasets, indicating their good cross-cultivar adaptability. In contrast, the RF and AdaBoost models exhibited significant performance degradation on the Murcott dataset (Table 8). Furthermore, the CatBoost model showed signs of overfitting. Across both the Mianju and Murcott datasets, the optimal model performance for HLB-infected leaves (Rv^2^ = 0.816 for Mianju; Rv^2^ = 0.730 for Murcott) was consistently and significantly lower than that for healthy leaves (Rv^2^ = 0.956 for Mianju; Rv^2^ = 0.882 for Murcott). This finding suggests that HLB infection alters the relationship between chlorophyll content and spectral features, making it more complex, nonlinear, and noisy than the relatively simple and stable relationship observed in healthy leaves. Consequently, even the best-performing model fails to achieve prediction accuracy comparable to that for healthy leaves.

### 4.3. Future Perspectives in Chlorophyll Estimation for HLB-Infected Citrus

In this study, the LCC of healthy and HLB-infected leaves was successfully estimated by investigating spectral changes in HLB-infected leaves and comparing the differences in chlorophyll feature bands between healthy leaves and HLB-infected leaves. Thus, a quantitative relationship was established between the hyperspectral features and the chlorophyll content induced by HLB. These findings reveal that the use of hyperspectral data to estimate chlorophyll content is not only an effective method for assessing citrus health but also a reliable indirect indicator for the early identification and monitoring of HLB, indicating significant application potential.

However, this study has several limitations: (1) while validation on the Murcott variety has been supplemented, the research primarily focused on LCC prediction for healthy and HLB-infected leaves during the fruit maturation stage of Mianju, without covering other citrus varieties or additional critical growth stages; (2) a larger dataset is needed to obtain more robust experimental results. Future studies should explore the application of deep learning and stacked ensemble models to increase their comprehensiveness and reliability. Future research should not merely pursue high accuracy on a single dataset but must treat cross-domain validation as a core component of model evaluation. Based on these limitations, future work should: (1) systematically construct large-scale hyperspectral datasets encompassing multiple varieties, production regions, and growth stages to establish a foundation for developing truly universal models; (2) explore advanced algorithms such as deep learning, transfer learning, and domain adaptation to enhance model adaptability to unfamiliar environments and varieties, thereby further strengthening the comprehensiveness and credibility of the research.

## 5. Conclusions

In this study, hyperspectral technology was combined with machine learning algorithms to select feature bands and construct corresponding LCC estimation models for both healthy and HLB-infected citrus leaves. By comparing the spectral characteristics and LCC estimation models for both healthy and HLB-infected citrus leaves, the correlation between HLB and LCC changes was thoroughly investigated. The main conclusions are presented as follows:(1)HLB-infected leaves exhibited significant spectral differences in the visible and red-edge regions, with average reflectance at 554 nm and 710 nm being 0.144 and 0.209 higher, respectively, compared to healthy leaves. The infected leaves also showed a distinct blue shift in the red-edge position.(2)Comparative analysis of the characteristic bands selected by LASSO and spectral indices revealed that the proportion of characteristic bands for HLB-infected leaves located in the NIR region was markedly greater than that for healthy leaves. Specifically, more than 66% of the bands in the SI_CB characteristic band for HLB-infected leaves were in the NIR region.(3)The modeling results based on six machine learning algorithms indicated that the feature bands selected by LASSO outperformed those derived from spectral indices for predicting citrus LCC. Among the different models, PLSR yielded the best performance, achieving an Rv^2^ of 0.956 for healthy leaves and an Rv^2^ of 0.816 for HLB-infected leaves, which confirms its high estimation accuracy and robustness.(4)According to the statistical table of LCC in sweet citrus leaves, the average LCC of healthy leaves in November was approximately 1.57 times that of HLB-infected leaves. By comparing the LCC estimation models, this study confirmed that the optimal model for healthy leaves outperformed that for HLB-infected leaves. Furthermore, quantitative relationships between the characteristic bands and chlorophyll content were established for both healthy and HLB-infected leaves.

## Figures and Tables

**Figure 1 sensors-25-07292-f001:**
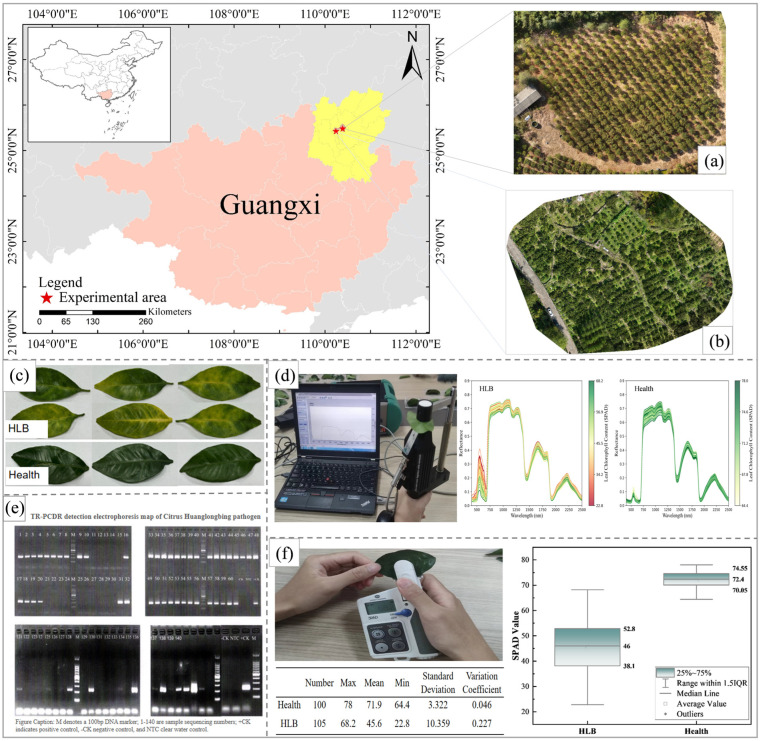
Overview of the study area and data collection: (**a**) the study area of the citrus experimental base in Sanjie town (**b**) the study area of the citrus experimental base in Tanxia town, (**c**) leaf samples, (**d**) hyperspectral imaging of leaves and spectral curves of HLB-infected and healthy leaves, (**e**) TR-PCDR detection report for the leaf samples, and (**f**) SPAD measurements of leaves and box plot comparison of SPAD values between HLB-infected and healthy leaves.

**Figure 2 sensors-25-07292-f002:**
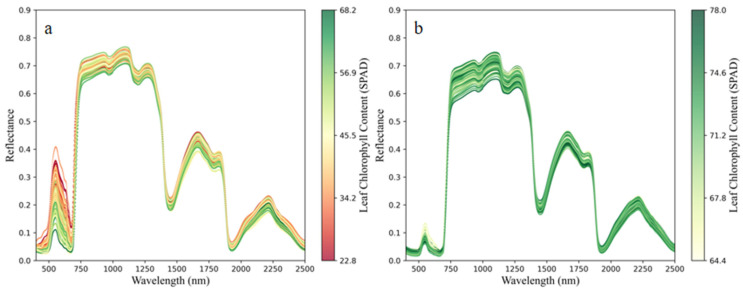
Color mapping of citrus LCC and hyperspectral data: (**a**) color mapping of HLB-infected Mianju citrus LCC and hyperspectral data and (**b**) color mapping of healthy citrus LCC and hyperspectral.

**Figure 3 sensors-25-07292-f003:**
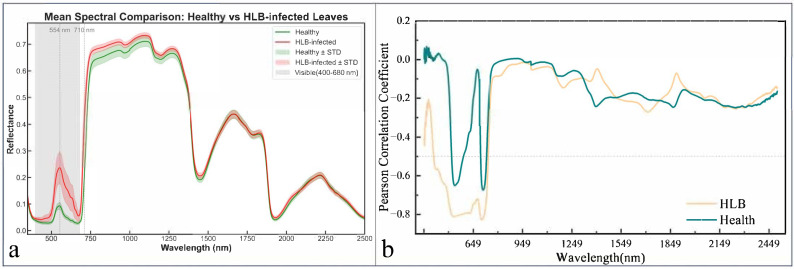
Mean spectral reflectance curves and pearson correlation coefficient between LCC and hyperspectral data. (**a**), Comparison of average hyperspectral reflectance curves between healthy and HLB-infected leaves; (**b**), Pearson correlation coefficient plot between SPAD values and hyperspectral reflectance for healthy and HLB-infected leaves.

**Figure 4 sensors-25-07292-f004:**
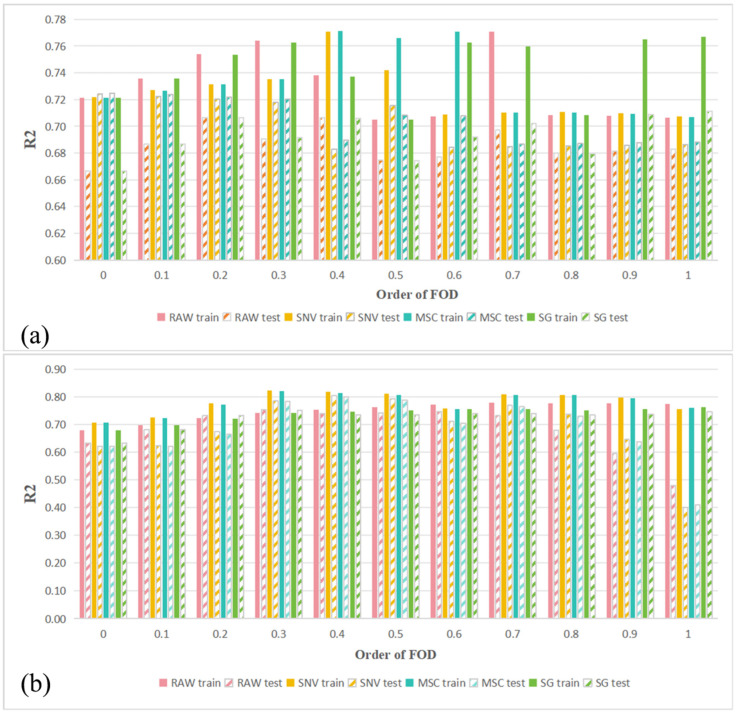
Performance evaluation of LCC estimation using the PLSR model with all the bands: (**a**) using the PLSR model to evaluate the effect of different spectral preprocessing combinations on LCC prediction for HLB-infected Mianju leaves and (**b**) using the PLSR model to evaluate the effect of different spectral preprocessing combinations on LCC prediction for healthy Mianju leaves.

**Figure 5 sensors-25-07292-f005:**
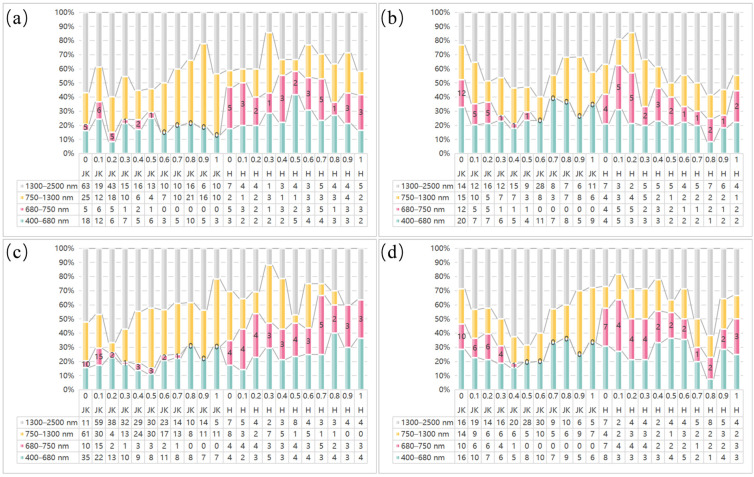
Proportion of feature bands selected by the LASSO algorithm across spectral regions with different preprocessing combinations: (**a**–**d**) represent the preprocessing combinations RAW_FOD, SNV_FOD, SG_FOD, and MSC_FOD in sequence; JK denotes healthy leaves, H denotes leaves infected with HLB, and 0–1 represents the order of the FOD.

**Figure 6 sensors-25-07292-f006:**
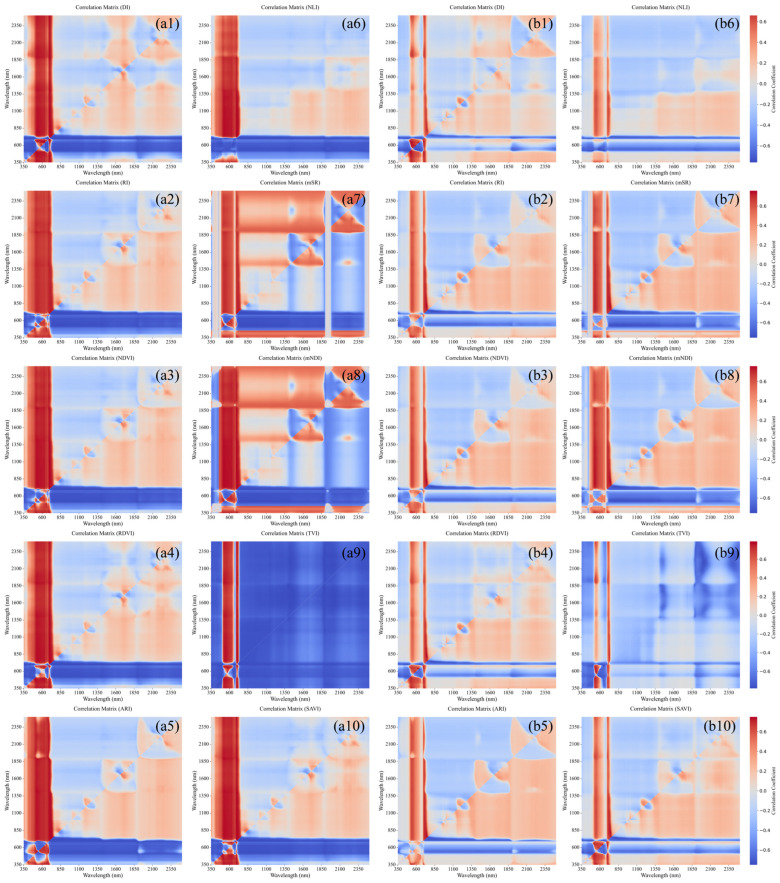
Correlation coefficient matrix between spectral indices and LCC for HLB-infected and healthy Mianju leaves. (**a1**–**a10**): correlation coefficient matrix between the spectral indices RI, DI, NDVI, RDVI, ARI, NLI, mSR, mNDI, TVI, SAVI and LCC for HLB-infected leaves; (**b1**–**b10**): correlation coefficient matrix between the spectral indices RI, DI, NDVI, RDVI, ARI, NLI, mSR, mNDI, TVI, SAVI and LCC for healthy leaves.

**Figure 7 sensors-25-07292-f007:**
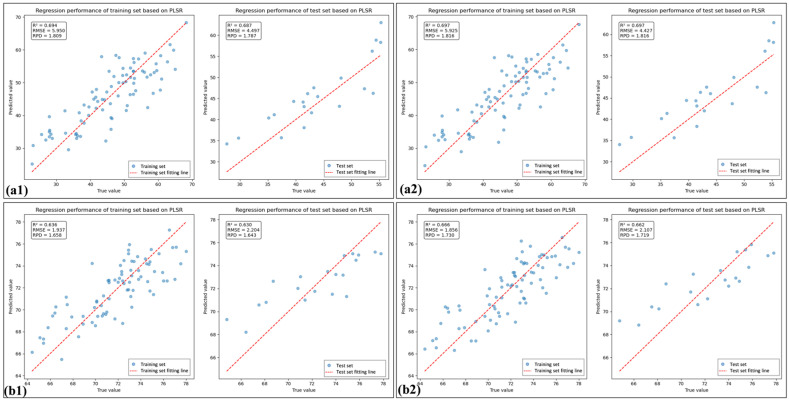
The linear fitting plots of spectral indices based on PLSR: (**a1**) and (**a2**) are linear fitting plots of 10 spectral indices and 8 spectral indices for HLB-infected leaves, respectively; (**b1**) and (**b2**) are linear fitting plots of 10 spectral indices and 6 spectral indices for healthy leaves, respectively.

**Figure 8 sensors-25-07292-f008:**
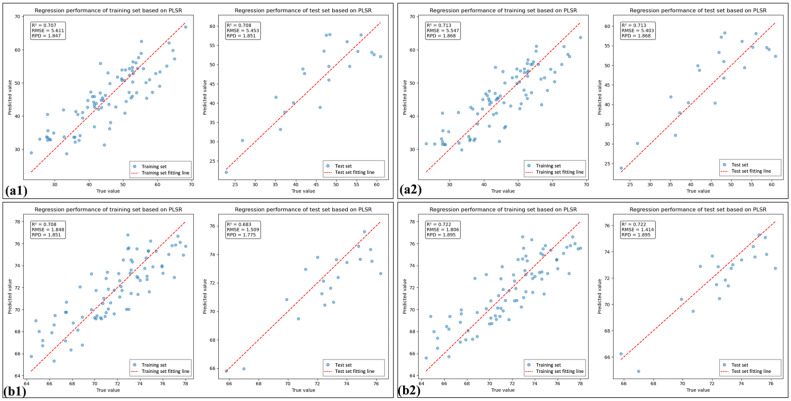
The linear fitting plots of feature bands selected using spectral indices based on PLSR: (**a1**) and (**a2**) are linear fitting plots of the SI_CB and SI_CB_PLS for HLB-infected leaves based on PLSR, respectively; (**b1**) and (**b2**) are linear fitting plots of the SI_CB and SI_CB_PLS feature for healthy leaves based on PLSR, respectively.

**Figure 9 sensors-25-07292-f009:**
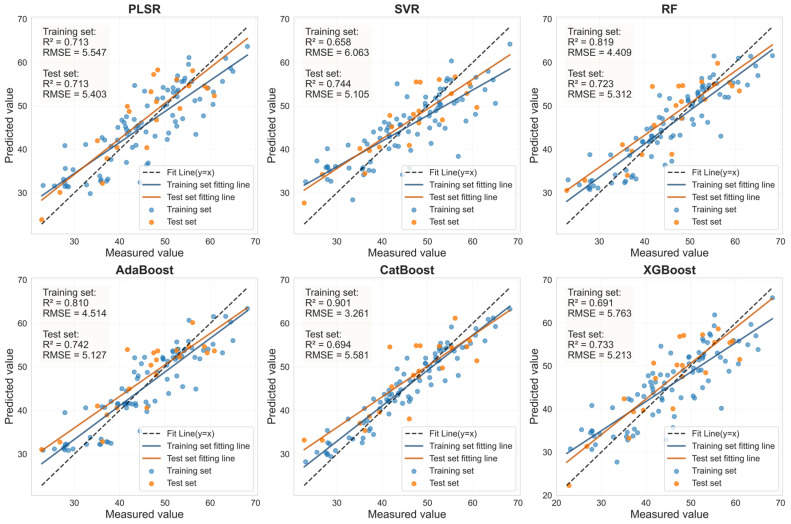
Evaluation of LCC prediction performance based on six machine learning models for the SI_CB_PLS feature bands in HLB-infected leaves.

**Figure 10 sensors-25-07292-f010:**
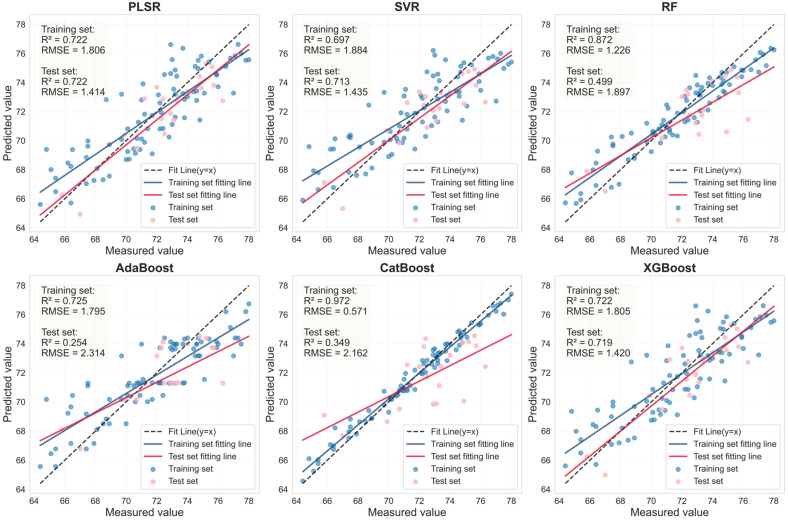
Evaluation of LCC prediction performance based on six machine learning models for the SI_CB_PLS feature bands in healthy leaves.

**Figure 11 sensors-25-07292-f011:**
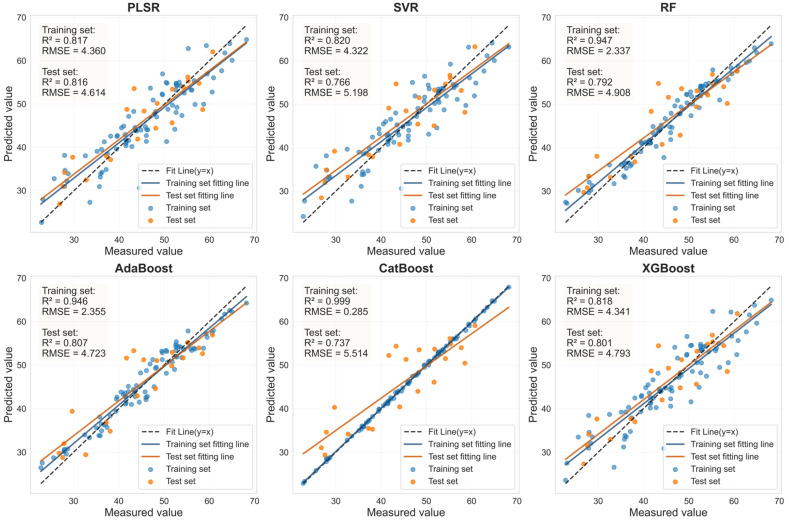
Evaluation of LCC prediction performance based on six machine learning models for the LASSO feature bands in HLB-infected leaves.

**Figure 12 sensors-25-07292-f012:**
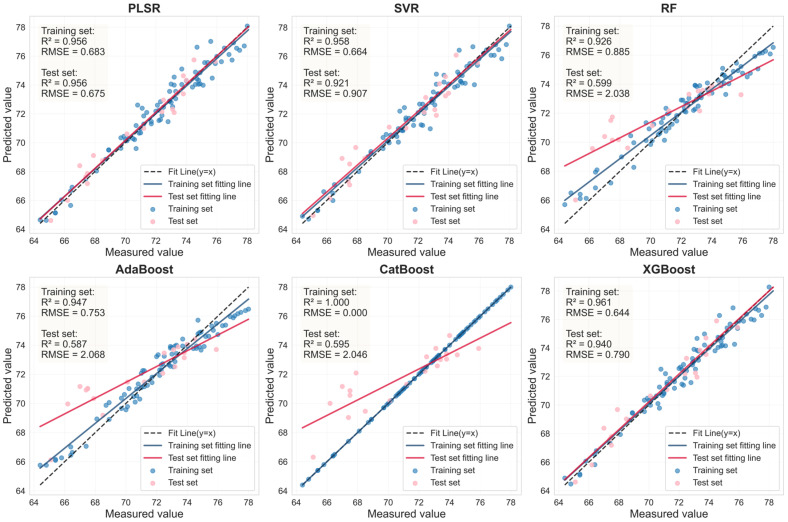
Evaluation of LCC prediction performance based on six machine learning models for the LASSO feature bands in healthy leaves.

**Figure 13 sensors-25-07292-f013:**
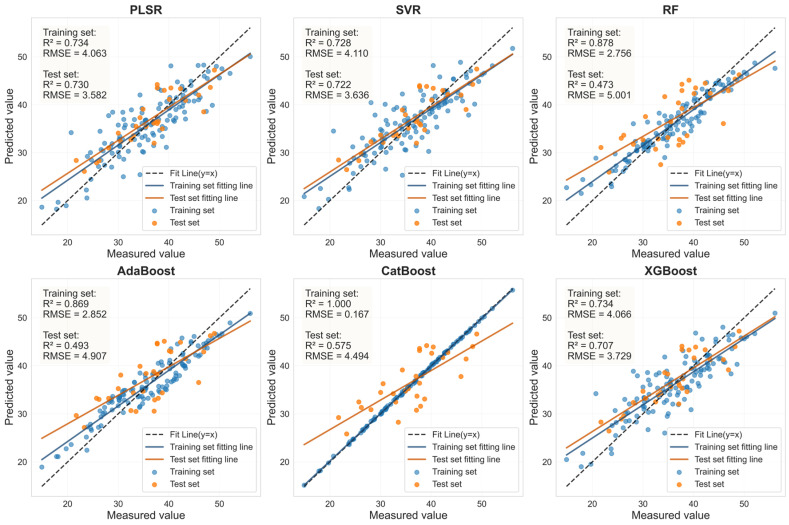
LASSO feature bands for Murcott HLB-infected leaves based on linear regression fitting plots from six machine learning models.

**Figure 14 sensors-25-07292-f014:**
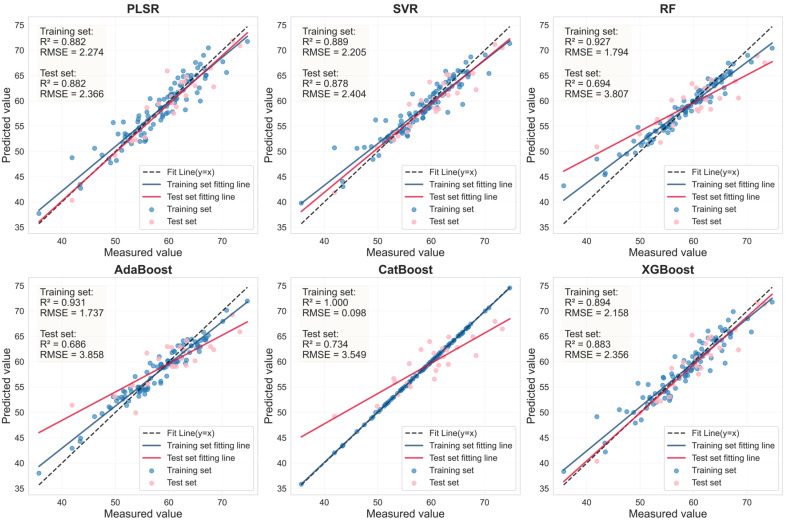
LASSO feature bands for Murcott health leaves based on linear regression fitting plots from six machine learning models.

**Table 1 sensors-25-07292-t001:** Statistical analysis of SPAD values in Mianju citrus leaves under different health conditions.

Health Status	Sample Size	Max	Mean	Min	Standard Deviation	Coefficient of Variation
Healthy	100	78	71.9	64.4	3.322	0.046
HLB-infected	105	68.2	45.6	22.8	10.359	0.227

**Table 2 sensors-25-07292-t002:** 10 Formulas for typical spectral indices.

Spectral Index	Formula	Reference
Ratio index (RI)	R_i_/R_j_	[47]
Difference index (DI)	R_i_ − R_j_	[47]
Normalized difference vegetation index (NDVI)	R_i_ − R_j_/R_i_ + R_j_	[47]
Renormalized difference vegetation index (RDVI)	(R_i_ − R_j_)/SQRT(R_i_ + R_j_)	[47]
Anth reflectance index (ARI)	(R_j_ − Rj)/(R_j_ × Rj)	[47]
Nonlinear vegetation index (NLI)	(R^2^_i_ − R_j_)/(R^2^_i_ + R_j_)	[47]
Modified simple ratio (mSR)	R_i_ − R_445_/R_j_ − R_445_	[46]
Modified normalized difference index (mNDI)	R_i_ − R_j_/R_i_ + R_j_ − 2R_445_	[46]
Triangular vegetation index (TVI)	0.5 × (120 × (R_i_ − R_500_)) − 200 × (R_j_ − R_500_)	[46]
Soil–adjusted vegetation index (SAVI)	$(1+0.16)\;\frac{R_{i}-R_{j}}{R_{i}+R_{j}+0.16}$	[48]

**Table 3 sensors-25-07292-t003:** Regression performance of LCC prediction using PLS with different feature band combinations.

Leaf Condition	Feature Selection	Pretreatment	FOD Order	Band Number	Rc^2^	RMSEc	Rv^2^	RMSEv
Health	LASSO	RAW	1	23	0.95	0.76	0.94	0.81
		SNV	1	28	0.92	0.93	0.90	1.00
		SG	0.8	34	0.92	0.94	0.93	0.88
		MSC	0.9	27	0.92	0.91	0.91	0.94
	RFE	RAW	0.8	30	0.85	1.27	0.81	1.40
		SNV	0.9	30	0.86	1.22	0.84	1.30
		SG	0.5	30	0.78	1.54	0.79	1.46
		MSC	0.8	30	0.84	1.29	0.85	1.26
	CARS	RAW	0.9	143	0.98	0.45	0.89	1.06
		SNV	0.6	190	0.91	0.97	0.90	1.04
		SG	0.7	123	0.94	0.77	0.88	1.13
		MSC	1	144	0.92	0.94	0.85	1.25
HLB	LASSO	RAW	0.9	14	0.82	4.34	0.82	4.70
		SNV	1	9	0.80	4.51	0.81	4.75
		SG	0.5	17	0.77	4.92	0.75	5.41
		MSC	1	12	0.81	4.44	0.78	5.00
	RFE	RAW	0.5	30	0.75	5.08	0.75	5.42
		SNV	0.8	30	0.74	5.17	0.74	5.48
		SG	0.1	30	0.75	5.11	0.73	5.63
		MSC	0.7	30	0.75	5.12	0.74	5.44
	CARS	RAW	0.1	81	0.68	5.79	0.68	6.07
		SNV	0.1	144	0.74	5.23	0.73	5.61
		SG	0.4	60	0.76	5.03	0.65	6.34
		MSC	0.2	61	0.75	5.07	0.71	5.79

**Table 4 sensors-25-07292-t004:** Optimal band combinations for 10 spectral indices and their corresponding maximum absolute correlation coefficients.

Spectral Index	HLB		Healthy	
	Band Combination	Correlation Coefficient	Band Combination	Correlation Coefficient
RI	(713, 712)	0.826	(820, 739)	0.747
DI	(830, 703)	0.841	(643, 529)	0.754
NDVI	(830, 699)	0.833	(819, 741)	0.747
RDVI	(714, 713)	0.838	(636, 631)	0.742
ARI	(1896, 689)	0.795	(751, 750)	0.736
NLI	(842, 699)	0.835	(352, 531)	0.659
mSR	(714, 713)	0.825	(814, 739)	0.757
mNDI	(830, 699)	0.835	(814, 739)	0.757
TVI	(842, 702)	0.835	(530, 537)	0.791
SAVI	(818, 700)	0.835	(636, 631)	0.751

**Table 5 sensors-25-07292-t005:** Performance evaluation of LCC prediction based on spectral index-derived feature variable combinations.

Sample	Feature Combination Name	Feature Variable	Rc^2^	RMSEc	Rv^2^	RMSEv
HLB	SI	RI, DI, NDVI, RDVI, ARI, NLI, mSR, mNDI, TVI, SAVI	0.694	5.950	0.687	4.497
	SI_PLS	DI, NDVI, RDVI, NLI, mSR, mNDI, TVI, SAVI	0.697	5.925	0.697	4.427
	SI_CB	689, 699, 700, 702, 703, 712, 713, 714, 818, 830, 842, 1896	0.707	5.611	0.708	5.453
	SI_CB_PLS	702, 703, 713, 714, 1896	0.713	5.547	0.713	5.403
Health	SI	RI, DI, NDVI, RDVI, ARI, NLI, mSR, mNDI, TVI, SAVI	0.636	1.937	0.630	2.204
	SI_PLS	DI, ARI, NLI, mSR, TVI, SAVI	0.666	1.856	0.662	2.107
	SI_CB	352,5 29, 530, 531, 537, 631, 636, 643, 739, 741, 750, 751, 814, 819, 820	0.708	1.848	0.683	1.509
	SI_CB_PLS	352, 529, 643, 739, 741, 750, 814, 819, 820	0.722	1.895	0.722	1.414

**Table 6 sensors-25-07292-t006:** Optimal band combination for HLB-infected and healthy leaves.

Sample	Feature Variable	Wavelength/nm
HLB	RAW-FOD0.9-LASSO	404, 409, 511, 720, 722, 725, 818, 840, 1077, 1245, 2319, 2320, 2239, 2399
	SI_CB_PLS	702, 703, 713, 714, 1896
Health	RAW-FOD1.0-LASSO	438, 558, 593, 751, 823, 863, 887, 899, 923, 953, 965, 1070, 1076, 1742, 1777, 1906, 2125, 2283, 2322, 2412, 2423, 2424, 2483
	SI_CB_PLS	352, 529, 643, 739, 741, 750, 814, 819, 820

**Table 7 sensors-25-07292-t007:** Performance evaluation of feature bands for LCC prediction based on machine learning models.

Sample	Feature Variable	Model	Number	Rc^2^	RMSEc	Rv^2^	RMSEv	RPD
Health	RAW- FOD1.0- LASSO	PLSR	23	0.956	0.683	0.956	0.675	4.767
		SVR	23	0.958	0.664	0.921	0.907	3.548
		RF	23	0.926	0.885	0.599	2.038	1.578
		AdaBoost	23	0.947	0.753	0.587	2.068	1.556
		CatBoost	23	1.000	0.000	0.595	2.046	1.572
		XGBoost	23	0.961	0.644	0.940	0.790	4.074
	SI_CB _PLS	PLSR	9	0.722	1.895	0.722	1.414	1.895
		SVR	9	0.697	1.884	0.713	1.435	1.867
		RF	9	0.872	1.226	0.499	1.897	1.412
		AdaBoost	9	0.725	1.795	0.254	2.314	1.158
		CatBoost	9	0.972	0.571	0.349	2.162	1.239
		XGBoost	9	0.722	1.805	0.719	1.420	1.886
HLB	RAW- FOD0.9- LASSO	PLSR	14	0.817	4.360	0.816	4.614	2.331
		SVR	14	0.820	4.322	0.766	5.198	2.069
		RF	14	0.947	2.337	0.792	4.908	2.191
		AdaBoost	14	0.946	2.355	0.807	4.723	2.277
		CatBoost	14	0.999	0.285	0.737	5.514	1.950
		XGBoost	14	0.818	4.341	0.801	4.793	2.244
	SI_CB _PLS	PLSR	5	0.713	5.547	0.713	5.403	1.868
		SVR	5	0.658	6.063	0.744	5.105	1.977
		RF	5	0.819	4.409	0.723	5.312	1.900
		AdaBoost	5	0.810	4.514	0.742	5.127	1.968
		CatBoost	5	0.901	3.261	0.694	5.581	1.808
		XGBoost	5	0.691	5.763	0.733	5.213	1.936

**Table 8 sensors-25-07292-t008:** Performance evaluation of LCC prediction using LASSO feature bands combined with six machine learning models on the Murcott.

Sample	Feature Variable	Num	Model	Rc^2^	RMSEc	Rv^2^	RMSEv	RPD
Health	RAW- FOD1.0- LASSO	44	PLSR	0.882	2.274	0.882	2.366	2.908
			SVR	0.889	2.205	0.878	2.404	2.862
			RF	0.927	1.794	0.694	3.807	1.807
			AdaBoost	0.931	1.737	0.686	3.858	1.783
			CatBoost	1.000	0.098	0.734	3.549	1.939
			XGBoost	0.894	2.158	0.883	2.356	2.920
HLB	RAW- FOD0.9- LASSO	34	PLSR	0.734	4.063	0.730	3.583	1.923
			SVR	0.728	4.110	0.722	3.636	1.895
			RF	0.878	2.756	0.473	5.001	1.378
			AdaBoost	0.869	2.852	0.493	4.907	1.404
			CatBoost	1.000	0.167	0.575	4.494	1.533
			XGBoost	0.734	4.066	0.707	3.729	1.848

## Data Availability

The original contributions presented in this study are included in the article. Further inquiries can be directed to the corresponding author.

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
