# Peer review of "Investigating the Association Between Citrus Huanglongbing and Chlorophyll Content Using Hyperspectral Detection"

_sensors, 2025, doi:10.3390/s25237292_

Round 1
Reviewer 1 Report
Comments and Suggestions for Authors
This paper investigates the relationship between citrus Huanglongbing (HLB) and chlorophyll content. By combining hyperspectral reflectance data, various feature selection algorithms, and machine learning models, the chlorophyll content of healthy and infected leaves was quantitatively estimated. This research topic has significant application value and is of great importance for the early diagnosis and quantitative spectral analysis of citrus diseases. However, the paper has certain shortcomings in logic, methodology, and expression, and the conclusions lack sufficient data support. It is recommended that the authors revise the paper comprehensively and thoroughly before resubmitting it for review.
(1)Figure 1. (d) and (e) contain Chinese labels, which violates the standard for English journal figures. We recommend that all labels be replaced with English.
(2)As shown in Figure 1(c), the authors collected hyperspectral data only from a local portion of the leaf rather than from the entire leaf surface. However, the description in the methodology section “hyperspectral data of leaf samples were collected” may mislead readers into assuming that the measurements covered the whole leaf. Since different parts of citrus leaves exhibit significant structural and pigment differences, measuring only a small region may reduce the representativeness of the spectral data. It is recommended that the authors clearly specify in Section 2.2 the exact measurement area and provide the scientific rationale for selecting that particular region.
(3)Section 2.3, concerning sample testing, only vaguely states that "HLB infection status is determined by qPCR testing," but fails to provide specific testing methods, resulting in a lack of methodological transparency. It is recommended that the authors supplement this with detailed qPCR technical specifications and related information.
(4)The article only divides the dataset into training and validation sets, lacking an independent test set. This may lead to overestimation of the model’s performance and make it difficult to reflect its generalization ability on unseen data. It is recommended that the authors include an independent test set to enhance the reliability of the model results.
(5)Figure 6 has insufficient resolution; the axis labels and content in the figure are blurry and difficult to read, which may affect the readers’ understanding of the results. It is recommended that the authors re-export a version with a resolution of ≥300 dpi.
Reviewer 2 Report
Comments and Suggestions for Authors
The authors of the manuscript attempted to establish a relationship between spectral characteristics and chlorophyll content in citrus leaves (LCC) infected with Citrus Huanglongbing (HLB) for the early detection of HLB in Mianju mandarin cultivars.
General comments:
It is known that HLB causes yellowing of shoots and leaves, general plant depression, growth retardation and desiccation. Why was it necessary to establish a connection only with chlorophyll, without taking into account other signs, in order to detect HLB? It should be noted that technologies that allow detecting affected plants before visible signs appear are of particular interest.
Chlorophyll concentration may decrease not only due to HLB, but also for many other reasons, such as light stress. How did you determine the cause of the decrease in chlorophyll based on spectral data?
In my opinion, the weak point is that the study is based on data obtained from sampling in a single study area in field conditions on 19 November 2024. It is well known that machine learning models often produce poor results when tested at other times or in other areas. It is necessary to test the models obtained at other sites at other times.
Please note that:
SPAD-502 – The device measures the spectral absorption of light by leaves at two wavelengths, 650 and 940. This data is used to indirectly determine the concentration of chlorophyll.
ASD FieldSpec 4 Stand Res – a spectral sensor with a spectral resolution of 1 nm and a range of 350-2500 nm.
No direct methods of chlorophyll measurement were used.
The study performed a regression analysis of data from one spectral device (SPAD-502) with another (FieldSpec 4), both devices having wavelengths for measurement of 650 and 940 nm.
There are no quantitative data in the abstract. Qualitative characteristics of the models under development should be avoided, and quantitative characteristics should be specified (e.g. ‘demonstrated high predictive power’, ‘optimal model's performance was superior’).
The figures need to be improved in terms of quality. In some figures, the font is too small, for example figures 6 and 12.
Specific comments:
L 22. ‘spectral indices’ – Please specify which vegetation indices were used.
L 22-23. ‘several machine learning model’ – Please specify which machine learning methods were used.
L 28. ‘high predictive power’ – Quantitative characteristics are required.
L 29-30. Key words should not duplicate terms in the manuscript title (e.g. ‘citrus’, ‘hyperspectral’, ‘Huanglongbing’).
L 53. ‘HLB decreased by 3 to 4 times’ – Please indicate at what stage of growth and how long after infection?
L 55-56. ‘Therefore, as a direct indicator of photosynthetic capacity, the precise estimation of chlorophyll content is essential for the early detection of HLB infection.’ Chlorophyll concentration can decrease for many other reasons, including light stress. How did you determine the cause of the decrease in chlorophyll based on spectral data?
L 70-71. ‘at the unmanned aerial vehicle (UAV) scale, canopy scale, and leaf scale’ -> Landscape, canopy and leaf scale.
L 74-77. References are required.
L 78-79. References are required.
L 84-85. References are required.
L 86-87. References are required.
L 88. ‘Not all feature selection methods, however, improve modelling performance.’ This is common knowledge.
L 89. ‘choosing the optimal feature selection algorithm is essential’. This is common knowledge.
L 96-97. ‘As mentioned above, significant spectral differences exist between HLB-infected leaves and healthy citrus leaves.’ This is an unnecessary sentence.
L 97-99. ‘Combining hyperspectral data with machine learning models to estimate the chlorophyll content in HLB-infected and healthy citrus leaves is feasible for early HLB identification’ If so, what is the scientific novelty of your research? References are required.
L 125. Fig 1. All captions in the figures must be in English.
L 146. ‘Savitzky–Golay smoothing’ Please specify the step.
L 156. ‘significant’ – Quantitative characteristics are required.
L 180. ‘Pearson correlation analysis’. Normal distribution of data is required to use parametric statistical methods. Please provide the results of the normality tests.
L 201. ‘typical spectral indices’. Why are these 10 vegetation indices typical? Please provide a reference. How were they selected? There are currently more than 200 vegetation indices. Why were other chlorophyll-sensitive vegetation indices not used?
L 205-209. It is unclear what this is about.
L 206. ‘determining the 10 optimal spectral indices’. Out of 10 typical spectral indices? That's not clear.
L 208, 238. ‘[Error! Reference source not found.]’. Please correct it.
L 537. ‘Significant differences in spectral reflectance were observed’. – Quantitative characteristics are required.
Round 2
Reviewer 1 Report
Comments and Suggestions for Authors
The authors have satisfactorily addressed all previous comments, and the manuscript has been significantly improved. I am satisfied with the current version and recommend the manuscript for publication.
Reviewer 2 Report
Comments and Suggestions for Authors
Dear authors!
Thank you for your detailed answers to my questions, the additional experiments you conducted, and the significant revisions to the manuscript. I believe the manuscript in its current form is suitable for publication.